# Posttraumatic Stress Disorder Mediates the Association between Traumatic World Trade Center Dust Cloud Exposure and Ongoing Systemic Inflammation in Community Members

**DOI:** 10.3390/ijerph19148622

**Published:** 2022-07-15

**Authors:** Yian Zhang, Rebecca Rosen, Joan Reibman, Yongzhao Shao

**Affiliations:** 1Department of Population Health, NYU Grossman School of Medicine, 180 Madison Avenue, New York, NY 10016, USA; yian.zhang@nyulangone.org; 2HHC World Trade Center Environmental Health Center, 462 First Avenue, New York, NY 10016, USA; rebecca.rosen@nychhc.org; 3NYU Alzheimer Disease Research Center, 145 E 32 Street, New York, NY 10016, USA; 4Department of Psychiatry, NYU Grossman School of Medicine, 550 First Avenue, New York, NY 10016, USA; 5Department of Medicine, NYU Grossman School of Medicine, 550 First Avenue, New York, NY 10016, USA

**Keywords:** C-reactive protein, systemic inflammation, PTSD symptom cluster, partial mediation, PCL score, cognitive impairment, WTC dust cloud

## Abstract

Exposure to World Trade Center (WTC) dust/fumes and traumas on 11 September 2001 has been reported as a risk factor for post-traumatic stress disorder (PTSD) and other mental/physical health symptoms in WTC-affected populations. Increased systemic inflammation and oxidative stress from the exposure and subsequent illnesses have been proposed as contributors to the underlying biological processes. Many blood-based biomarkers of systemic inflammation, including C-reactive protein (CRP), are useful for non-invasive diagnostic and monitoring of disease process, and also potential targets for therapeutic interventions. Twenty years after 9/11, however, the relationships between WTC exposure, chronic PTSD, and systemic inflammation are only beginning to be systematically investigated in the WTC-affected civilian population despite the fact that symptoms of PTSD and systemic inflammation are still common and persistent. This paper aims to address this knowledge gap, using enrollees of the WTC Environmental Health Center (EHC), a federally designated treatment and surveillance program for community members (WTC Survivors) exposed to the 9/11 terrorist attack. We conducted a mediation analysis to investigate the association between acute WTC dust cloud traumatic exposure (WDCTE) on 9/11, chronic PTSD symptoms, and levels of systemic inflammation. The data indicate that the chronic PTSD symptoms and some specific symptom clusters of PTSD significantly mediate the WDCTE on systemic inflammation, as reflected by the CRP levels. As both chronic PTSD and systemic inflammation are long-term risk factors for neurodegeneration and cognitive decline, further research on the implications of this finding is warranted.

## 1. Introduction

The collapse of the World Trade Center (WTC) towers on 11 September 2001 due to a terrorist attack resulted in an enormous environmental disaster and negatively impacted a large number of local community members, including local residents, local workers, students, and those passing by the area on 9/11. Many community members experienced acute physical and psychological exposures, such as inhalation of dust from the dust clouds that were created as the WTC towers collapsed, and fear for their lives as they ran from and were engulfed by those massive dust clouds. Other acute psychological exposures included witnessing destruction, death, dismemberment, and escaping from the collapsing towers. For easy exposition, we subsequently refer to the WTC dust cloud traumatic exposure on 11 September 2001 as WDCTE. Chronic exposures included those from re-suspended dust and fumes, and witnessing months of rescue and recovery efforts, and displacement from work and homes [1,2,3,4]. In short, acute or chronic 9/11-related exposures for the WTC Survivors, including children and pregnant women, can be quite substantial. In fact, even among the children in the WTC-affected community, 12 years after 9/11, we and others have identified increased serum dioxins and furans in those who experienced WTC dust at home [5,6,7,8,9,10,11,12,13,14]. Well-described adverse health effects in the WTC disaster-exposed survivor and first responder populations include persistent aerodigestive disorders [15,16,17,18,19,20,21,22,23,24,25], cancers [26,27,28,29,30,31,32], and a high rate of post-traumatic stress disorder (PTSD) [16,19,24,33,34,35,36,37,38,39].

The WTC Environmental Health Center (WTC EHC) was established to treat and monitor community members’ health conditions that resulted from exposure to the terrorist attack and its aftermath [40]. The WTC EHC is a CDC/NIOSH-designated Center of Excellence in the WTC Health Program (WTCHP). All of the participants in the WTC EHC are required to have at least one CDC/NIOSH-certified mental health or medical condition related to specific WTC exposures. Initial and monitoring evaluations include assessments of both physical and mental health symptoms, including PTSD Checklist (PCL) scores. Based on these data, we previously reported high rates and persistence of PTSD symptoms in the WTC EHC population [37]. The high rates may be in part due to the diversity of the population, including many with low income and education level, factors previously reported to be associated with the development of PTSD after traumatic exposures [41,42]. Indeed, compared to the general WTC Responders, who are predominantly white male professionals, the members of the WTC EHC are civilians untrained for disasters, with an equal gender distribution, representation of diverse age, races, and ethnicities, as well as a wide range of social economic statuses. In addition, the presence of co-morbid mental–physical symptoms, including depression, anxiety, lower respiratory symptoms, and cancers, have also been associated with probable PTSD, reinforcing the need to consider co-morbid presentations [16,39,43].

The environmental exposures from WTC dust and fumes, as well as from general air pollution have been linked to a wide range of adverse health effects [2,44], including increased risk of mental health disorders [45,46,47], respiratory and cardiovascular diseases [14,23,48,49,50,51,52,53], neurotoxicity and neuropathy [54,55,56], cancers [26,27,28,29,30,31,32], and cognitive decline [57,58]. The biological mechanisms contributing to this wide range of adverse health effects have been hypothesized to involve systemic inflammation resulting from oxidative stress [51,59,60]. Indeed, measures of the inflammatory markers in the blood, including C-reactive protein (CRP) [61,62,63,64,65,66], white blood cell (WBC) counts [67], and Interleukin 6 (IL-6) [68], were shown to be elevated in populations with high particulate air pollution exposure. Importantly, most of the community members who were caught in the WTC dust cloud on 9/11 also had a traumatic life-threatening experience, thus, we and others have reported that elevated levels of CRP were associated with symptoms of PTSD and depression among the members of the WTC EHC [34,69,70,71]. Moreover, some studies further provide evidence that PTSD may induce systemic inflammation [72,73,74,75]. The ongoing systemic inflammation (observed many years after 9/11) is unlikely to be triggered directly by the original WDCTE, given the known short half-life of CRP. Therefore, we set out to test the alternative hypothesis that the chronic PTSD symptoms actually can induce systemic inflammation, by assessing whether the chronic PTSD symptoms mediate the effects of the WDCTE on the elevated CRP. Furthermore, the heterogeneous symptoms of PTSD can generally be clustered into four categories reflecting diverse components (re-experiencing; avoidance; negative alterations in cognitions and mood; and alterations in arousal and reactivity) [34,76,77]. We previously established that re-experiencing is significantly associated with elevated CRP levels among the members of the WTC EHC [34]. To extend our previous findings, this current paper further examines the differences in the effects of the four PTSD symptom clusters on mediating the association of WDCTE on the CRP level. This relationship might have further implications for aging and other long term health issues at the WTC EHC. In particular, PTSD is a well-known risk factor of cognitive decline [57,58,78,79,80], and also found as a mediator of the association between WTC exposure and subjective concerns of cognitive decline among the WTC-exposed firefighters [81]. Cognitive decline beyond normal aging has become a significant health concern in the aging 9/11 cohorts. Importantly, twenty years after 9/11, more than 75% of members of the WTC EHC are now more than 55 years old. Thus, in the long run, it is of interest to better understand the relationship between WTC dust cloud exposure, systemic inflammation, chronic PTSD symptoms, and cognitive decline in the WTC EHC for the affected community members. 

## 2. Methods 

### 2.1. Study Subjects

The patients from the WTC EHC at Bellevue Hospital Center with information on the CRP measures were included in the current study. The WTC EHC is a federally designated treatment and monitoring program for WTC Survivors including local workers, local residents, students, and those passing by the area of WTC on 9/11. All of the enrollees of the WTC EHC are required to have CDC/NIOSH-certified psychiatric or medical conditions related to specific WTC exposures. The inclusion criteria for patients to enroll in the WTC EHC were previously reported [82], and include presence as a local worker, resident, student or passer-by in the disaster area on 9/11, or presence as a local worker, resident, student between 11 September 2001 and 31 July 2002. The criteria were codified by the WTCHP and can be found at https://www.cdc.gov/wtc/eligiblegroups.html#nycSurvivor. CRP measurements were conducted for the patients at the initial or monitoring visits between August 2007 and January 2018. We included all of the patients with CRP data at the WTC EHC Bellevue Hospital and complete exposure information, and physical and mental health records. We excluded the patients who did not have CRP measurements in that time period. All of the data were recorded in the Institutional Review Board of New York University Grossman School of Medicine’s approved research database (NCT00404898), and only data from the patients who signed informed consent forms were utilized for analysis. The exposure information includes WTC dust cloud and exposure category classification [83], and the physical and mental health data include persistent lower respiratory symptoms (LRS), and symptoms of PTSD, depression, and anxiety [15,16,17,19,20,33,34,37]. A total of 731 patients from the WTC EHC were included in our analysis.

### 2.2. WTC Exposures and Medical Assessment

The patients completed a comprehensive, interviewer-administered questionnaire upon enrollment in the WTC EHC, that included demographic information and characterizations of WTC-related exposure [22]. Individuals who reported having been in the blinding dust cloud caused by the collapse of WTC buildings on 11 September 2001 were classified as having WTC dust cloud traumatic exposure. The potential for WTC acute/chronic exposures was also characterized by four categories: local worker; local resident (resident); clean-up worker; and other. The patients who reported more than one pack-year history of tobacco use were defined as ever smokers. The body mass index (BMI) of the patients was calculated, using information gathered during the initial medical visit. The presence and severity of LRS, of cough, wheeze, chest tightness, and dyspnea at rest, were measured by standardized health questionnaires [22]. The patients with symptoms more than twice per week during the month preceding enrollment were considered as persistent LRS.

### 2.3. Markers of Systemic Inflammation

A wide-range CRP (wr-CRP) assay (Siemen’s Diagnostic Center, Tarrytown, NY, USA) was used to measure the CRP. The wr-CRP assay is a clinically used measurement, with a wide range of sensitivity and a lower limit of detection of 0.12 mg/L. The wr-CRP correlates significantly with the high sensitivity CRP (hs-CRP) measurements and quantitation of microinflammatory activity in individuals [84]. A value > 3 mg/L was considered to be “High” [85]. The white blood cell (WBC) counts (per 10^3^ cells/mL) were also obtained for the patients from the lab studies, in the electronic medical records for the initial visit [86].

### 2.4. Mental Health Symptoms

PTSD symptom presence and severity were measured by the Post-traumatic Checklist-17 (PCL) [87]. Designation as positive for probable PTSD was defined as a PCL score ≥ 44 [19]. The questions from the PCL were also matched to the DSM-5 diagnostic criteria for characterization into four clusters, as previously described [34], reflecting symptoms of re-experiencing, avoidance, negative cognition/mood, and arousal. An average score for each patient was calculated for each cluster, ranging between 1–5. 

The Hopkins Symptom Checklist (HSCL-25) was used for detecting depression and anxiety [88]. These scales provided scores of depression (HSCL-D) and anxiety (HSCL-A) severity, where a score ≥ 1.75 is considered to suggest probable depression or probable anxiety.

### 2.5. Statistical Methods and Mediation Analyses

The descriptive statistics were calculated for systemic inflammation biomarkers (e.g., CRP and WBC count), demographic characteristics, exposures, LRS, and mental health symptoms, in which the continuous variables were summarized using mean and standard deviation (SD) or median and interquartile range (IQR) depending on the normality of the data. The categorical variables were summarized using count and percentage. The difference between the independent groups was assessed by two-sample *t*-tests or Mann–Whitney tests for continuous variables and by Chi-squared test for categorical variables.

To investigate whether PTSD (measured by PCL score) mediates the effect of WDCTE on systemic inflammation (measured by CRP level), we first performed mediation analysis using the regression method [89,90,91]. Figure 1 presents the basic concept of the general approach. Let *n* be the total number of study participants for i=1,⋯, n and denote *W*, *C*, *P*, and *V* as WDCTE, CRP level, PCL score, and some vector of other relevant covariates, respectively. Three multiple linear regression models are given as follows to assess the potential mediation effects: (1)$$C_{i}=\beta_{01}+\beta_{c}W_{i}+\beta_{v1}^{\prime }V_{i},$$
(2)$$P_{i}=\beta_{02}+\beta_{a}W_{i}+\beta_{v2}^{\prime }V_{i},$$
(3)$$C_{i}=\beta_{03}+\beta_{c^{\prime }}W_{i}+\beta_{b}P_{i}+\beta_{v3}^{\prime }V_{i},$$
where β denotes the regression coefficients. Model (1) assesses whether the WDCTE affects the CRP level with adjustment for relevant covariates *V*. $\beta_{c}$ (i.e., path c in Figure 1) is called *total effect* of WDCTE on CRP level. A significant *total effect* ($\beta_{c}$) suggests a further step to assess whether the PCL score is a mediator of the association between the WDCTE and CRP level. Model (2) regards the effect of the WDCTE on the PCL score (i.e., $\beta_{a}$, the path a in Figure 1). If $\beta_{a}$ is significant, the PCL score could be a potential mediator and Model (3) can be conducted. In Model (3), both the WDCTE and PCL scores are treated as predictors of the CRP level. To establish that the PCL score mediates the effect of the WDCTE on the CRP level, the effect of the PCL score on the CRP level after controlling for the WDCTE (i.e., $\beta_{b}$, path b in Figure 1) needs to be significant. The strength of mediation further depends on the effect size of the WDCTE on the CRP level, after controlling for the PCL score (i.e., $\beta_{c\prime }$, the path c’ in Figure 1, also called *direct effect*) and its significance. Although the conventional regression method for mediation analysis, as proposed by Baron and Kenny in 1986, used zero and non-zero effects instead of requiring statistical significance to determine the associations, in the above models, due to the concern of a small sample size [89], we used statistical significance in our case, in order to obtain more conclusive results. The causal variable WDCTE used in the above models is a binary variable (Yes/No). The CRP was log-transformed due to the skewed distribution, and the PCL score was used to reflect PTSD symptoms. All of the above models were adjusted for a vector of covariates *V*, including demographic variables (i.e., gender, race/ethnicity, age on 9/11, education, income, BMI, and smoking history), LRS (i.e., cough, wheeze, chest tightness, and dyspnea at rest), and WTC exposure category. To further obtain the amount of *mediation effect* or *indirect effect*, we used the formula $\beta_{c}-\beta_{c^{\prime }}=\beta_{a}\beta_{b}$ [89]. The 95% confidence interval (CI) and the related *p*-value of the *mediation effect* were calculated by a quasi-Bayesian Monte Carlo method, based on normal approximation [92]. We implemented this method using the “mediation” R package [93]. The same approaches were applied for the investigation of the mediation effect of each PTSD symptom cluster in the association between WDCTE and CRP levels. A value of *p* < 0.05 was used to test for two-sided statistical significance. All of the statistical analyses were conducted using R, version 4.1.3 (R Core Team, Vienna, Austria).

## 3. Results

### 3.1. Patient Characteristics

Table 1 presents the characteristics of the study group, which included 731 patients. Similar to our previous studies [22,34,94], half of the diverse population was female, 40% identified as Hispanic, the overall mean age on 9/11 was 43, about 66% had greater than high school education, and the majority were non-smokers and reported a low annual individual income (≤$30K). Half of the population reported being exposed to WDCTE, and the majority were local workers. The patients reported high proportions of persistent LRS and overall high positive rates of mental health symptoms, in which 43% had probable PTSD, 58% had depression, and 51% had anxiety. High CRP levels were also found, with levels > 3 mg/L in 37%. The median WBC count was reported as 6900 cells/mL with an IQR between 5700 cells/mL and 8300 cells/mL.

### 3.2. Characteristics Associated with PTSD Symptoms at Initial Visit 

Demographic characteristics, WTC exposures, LRS symptoms, CRP, and WBC count were univariately compared between the PTSD (PCL ≥ 44) and non-PTSD (PCL < 44) groups (Table 2). Similar to what we found before, the community members who self-reported as Hispanic (*p* = 0.002), had ≤ high school education (*p* = 0.002), or had income ≤ $30,000/year (*p* = 0.001), were more likely to have probable PTSD. The presence of any LRS symptoms (cough: *p* = 0.011; wheeze: *p* = 0.002; chest tightness: *p* < 0.001; and dyspnea at rest: *p* < 0.001) was related to a positive score for PTSD. The patients who self-reported as cleanup workers (*p* = 0.04) or being caught in the WTC dust cloud (*p* = 0.006) were also more likely to have PTSD symptoms. We also found significant elevations of CRP levels in the PTSD group compared to the non-PTSD group (*p* = 0.004). Another inflammatory marker, WBC count, also showed increased levels in the PTSD group.

### 3.3. PTSD Symptoms (PCL Score) Mediates the Association between WDCTE and Systemic Inflammation (CRP Level)

In this section, we report the mediation effect of the PTSD (PCL score) on the effect of WDCTE on systemic inflammation, as reflected in the CRP level. The regression-based method for mediation analysis has been described in the Statistical Methods and Mediation Analyses section, with Figure 1. Table 3 presents the results of the three regression models used for establishing the mediations and the summary of the mediation effect. Model (1) reflects the total effect of the WDCTE on the CRP level ($\beta_{c}$) after adjusting for demographic variables and LRS. The WDCTE is significantly related to elevated CRP levels ($\beta_{c}$ = 0.27, *p* = 0.02). In addition, the WDCTE is also significantly associated to the PCL score ($\beta_{a}$ = 5.06, *p* < 0.01), with adjustment of the other factors according to Model (2). The significances of $\beta_{c}$ and $\beta_{a}$ imply that PTSD, as measured by the PCL score, may mediate the effect of the WDCTE on the CRP level. In Model (3), the PCL score was still significantly associated with CRP level, while controlling for the WDCTE and adjusting for other factors ($\beta_{b}$ = 0.01, *p* = 0.01), but the WDCTE is no longer significantly associated with the CRP level while controlling for the PCL scores ($\beta_{c^{\prime }}$ = 0.22, *p* = 0.06). The effect of the WDCTE on the CRP levels decreased from $\beta_{c}$ = 0.27 of Model (1) to $\beta_{c^{\prime }}$ = 0.22 of Model (3), which suggests that the total effect of the WDCTE on the CRP level was strongly mediated by the PCL score. The mediation effect or indirect effect of the PCL score was computed as $\beta_{c}-\beta_{c^{\prime }}$ = 0.27 − 0.22 = 0.05, with the 95% CI ranging from 0.01 to 0.09 and *p* = 0.01, which yields about 17% of total effect of the WDCTE on the CRP level being mediated by PCL score. 

### 3.4. Mediation Effects of PTSD Symptom Clusters in the Association between WDCTE and Systemic Inflammation 

Table 4 shows the results of the mediation effects of each PTSD symptom cluster (sub-PCL score) in the association between the WDCTE and systemic inflammation as measured by the CRP level. The impact of the WDCTE on the CRP level was significantly mediated by re-experiencing (β = 0.05, *p* < 0.01 and 17.5% of the total effect of WDCTE on CRP level were mediated), avoidance (β = 0.03, *p* = 0.02), and negative cognition/mood symptoms (β = 0.04, *p* = 0.02).

## 4. Discussion 

The current study assessed the relationships between chronic PTSD symptoms and specific PTSD symptom clusters and ongoing systemic inflammation, as measured by the CRP level in the context of the traumatic WTC dust cloud exposure. The WTC Survivors studied were enrolled in the WTC Environmental Health Center (WTC EHC) and many of the individuals had acute traumatic exposures to the initial dust clouds created as the WTC buildings collapsed (WTC dust cloud) on 11 September 2001, as well as having witnessed death and dismemberment, and often experienced their own fear of death as they escaped collapsing buildings or were engulfed in blinding dust clouds. The adverse health effects included chronic symptoms of PTSD, depression, anxiety, upper and lower respiratory symptoms, cancers and cognitive decline [15,16,17,18,19,20,22,26,27,28,29,30,31,33,34,35,36,37,40,57,58]. 

Our findings enhance and expand upon our previous studies, suggesting an important association between the traumatic WTC dust cloud exposure, systemic inflammation, and post-traumatic stress pathology and PTSD symptom clusters [34]. In our study cohort, persistent symptoms of PTSD and depression have emerged as two highly prevalent and comorbid post-traumatic stress responses to the WTC disaster exposures. The existing literature on air pollution studies indicates that air pollution can lead to systemic inflammation and elevated CRP [61,62,63,64,65,66], and systemic inflammation has emerged in the literature as one of the plausible biological mechanisms in the pathogenesis of depression and PTSD [95,96]. Note that we have previously reported that most of the subjects with symptoms of PTSD experience chronic or persistent symptoms [37]. The possibility exists that the physiologic corollary to the re-experiencing cluster of PTSD symptoms is a mechanism that may actually trigger systemic inflammation responses, including CRP and acute inflammatory cytokines. Indeed, the elevated CRP levels have been reported in association with the chronic re-experiencing symptoms of PTSD after traumatic exposures [72,73,97], and with arousal symptoms [98]. Given the short half-life of CRP, the observed elevated CRP many years after 9/11 in these subjects with PTSD symptoms might be triggered by the re-experiencing symptoms of PTSD. We thus conducted further mediation analysis and found that chronic PTSD symptoms and PTSD symptom clusters were significant mediators of the traumatic WTC dust cloud exposure on the elevated levels of CRP. 

Newly published studies in the WTC firefighter first-responder cohort suggest that PTSD completely mediates the subjective cognitive complaints in WTC-affected first responders [81]. Our data are consistent with this report and suggest that chronic PTSD symptoms similarly mediate the ongoing systemic inflammation, as measured in CRP. Thus, persistent PTSD symptoms are accompanied by chronic systemic inflammation, and treatment of the PTSD may reduce the inflammation. Chronic systemic inflammation and PTSD are known risk factors for neuroinflammation, neurodegeneration, and cognitive decline [58]. Thus, a better understanding of the roles of systemic inflammation and of how to reduce this inflammation would have a significant health impact on the cognitive issues for the patients in the WTC EHC, and generally for WTC Survivors. This gains increasing importance, since more than 75% of the WTC Survivors in the WTC EHC are now over 55 years old, and cognitive decline is thus an increasing health issue for this group.

The current study has multiple limitations. First, the study participants were selected based on the inclusion criteria stated in the Methods section which are, in turn, based on the available patient information from existing clinical databases. Additional exclusion criteria may be helpful for future studies, for example, including the presence of co-existing inflammatory disease before 9/11 and/or those who had traumatic experiences other than WTC exposure. Second, both the levels of the CRP and the symptoms of PTSD are cross-sectional data. Longitudinal data would allow more formal causal inference. We are currently planning longitudinal studies to further investigate and validate these preliminary findings on the relationship between the symptoms of PTSD and the levels of serum CRP. This study used patients with available CRP levels measured for evaluation purposes at the WTC EHC during a limited time frame. Third, we adjusted for co-morbid LRS and obesity, but future studies should consider randomly sampling patients and then obtain levels of systemic inflammation markers in order to reduce the impact of comorbidities. Respiratory conditions and obesity are not the only types of WTC-related comorbidities associated with inflammation; it is desirable to adjust for other possible WTC-related co-morbidities in future studies. Additionally, the serum CRP is known to lack specificity as a measure of systemic inflammation and may be upregulated by additional co-morbid conditions. Our previous study evaluated the presence of co-morbid LRS, reduced spirometry, and increased forced oscillation measurements. Our findings highlight the importance of understanding the interaction between the traumatic exposures, PTSD, and inflammation, and reinforce the need for future studies including those involving investigation of this hypothesis via random sampling of patients and by including measures of additional blood-based biomarkers, such as IL-6 and other cytokines [96,99,100]. 

## 5. Conclusions

The goal of this study was to understand the relationship between the WTC exposures, chronic PTSD symptoms with specific symptom clusters, and ongoing systemic inflammation in a cohort of community members with acute and traumatic WTC exposures. The identification of chronic PTSD symptom clusters as potential mediators of exposure on ongoing systemic inflammation biomarkers may have long-term implications for the health of those exposed to the WTC disaster, especially as the population ages. Specifically, the identification of PTSD clusters as mediators of inflammation suggest potential treatment targets to reduce inflammation. Moreover, this finding is salient relative to the risk of cognitive decline in the WTC-affected community members, as chronic systemic inflammation is a known risk factor of neuroinflammation, which, in turn, can be a risk factor for neurodegeneration and cognitive decline [58]. The current study is limited by the available cross-sectional CRP readings collected for the purpose of treatment; thus, well-designed longitudinal studies of a more comprehensive set of blood biomarkers, to further investigate the potential causal relationships, are warranted. 

## Figures and Tables

**Figure 1 ijerph-19-08622-f001:**
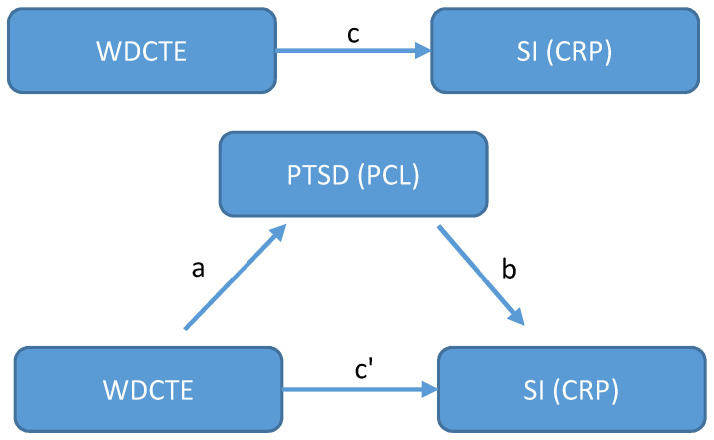
Diagrams for mediation analysis. We consider the effect of WTC dust cloud traumatic exposure (WDCTE) on systemic inflammation (SI), i.e., CRP level, is potentially mediated by PTSD (PCL score). Path c and c′ are the *total effect* and *direct effect* of WDCTE on CRP level; path a is the effect of WDCTE on PTSD (PCL); path b is the effect of PTSD (PCL) on CRP level.

**Table 1 ijerph-19-08622-t001:** Patient characteristics (*n* = 731).

Demographic Characteristics	
Gender, *n* (%)	
Female	373 (51.0)
Male	358 (49.0)
Age on 9/11 in year, mean (SD) ^a^	42.8 (11.5)
Race/ethnicity, *n* (%)	
Hispanic	289 (39.5)
Non-Hispanic white	221 (30.2)
Non-Hispanic Black	143 (19.6)
Other	78 (10.7)
Education, *n* (%)	
≤High school	252 (34.5)
>High school	479 (65.5)
Income, *n* (%)	
≤$30,000/year	465 (63.6)
>$30,000/year	266 (36.4)
BMI, mean (SD)	28.54 (6.2)
Ever smoker, *n* (%)	
Yes	230 (31.5)
No	501 (68.5)
**WTC Exposures**	
WTC dust cloud traumatic exposure, *n* (%)	
Yes	368 (50.3)
No	363 (49.7)
WTC exposure category classification, *n* (%)	
Local Worker	372 (50.9)
Resident	130 (17.8)
Clean-up worker	138 (18.9)
Other	91 (12.5)
**Lower respiratory symptoms within a month prior to enrollment**	
Cough, *n* (%)	
Yes	499 (68.2)
No	232 (31.7)
Wheezing, *n* (%)	
Yes	378 (51.7)
No	353 (48.3)
Chest tightness, *n* (%)	
Yes	463 (63.3)
No	268 (36.7)
Dyspnea at rest, *n* (%)	
Yes	296 (40.5)
No	435 (59.5)
**Positive mental health score**	
PTSD, *n* (%)	
Yes (PCL ≥ 44)	316 (43.2)
No (PCL < 44)	415 (56.8)
Depression, *n* (%)	
Yes (HSCL-D ≥ 1.75)	427 (58.4)
No (HSCL-D < 1.75)	304 (41.6)
Anxiety, *n* (%)	
Yes (HSCL-A ≥ 1.75)	370 (50.6)
No (HSCL-A < 1.75)	361 (49.4)
Any of above mental health issues, *n* (%)	
Yes	476 (65.1)
No	255 (34.9)
**CRP**	
CRP in mg/L, median [IQR] ^b^	1.6 [0.4, 5.3]
CRP > 3 mg/L, *n* (%)	
Yes	269 (36.8)
No	462 (63.2)
**WBC**	
WBC count in 10^3^ cells/mL, median [IQR]	6.9 [5.7, 8.3]

^a^ SD, standard deviation; ^b^ IQR, inter-quartile range.

**Table 2 ijerph-19-08622-t002:** Univariate analyses for association of PTSD with each of the predictors (*n* = 731).

	PTSD		*p*-Value ^c^
	No (PCL < 44) *n* = 415	Yes (PCL ≥ 44) *n* = 316	
**Demographics**			
Gender, *n* (%)			0.432
Female	206 (49.6)	167 (52.8)	
Male	209 (50.4)	149 (47.2)	
Age on 911 in year, mean (SD) ^a^	42.9 (12.6)	42.6 (9.7)	0.725
Race/ethnicity, *n* (%)			**0.002**
Hispanic	139 (33.5)	150 (47.5)	
Non-Hispanic white	139 (33.5)	82 (25.9)	
Non-Hispanic Black	91 (21.9)	52 (16.5)	
Other	46 (11.1)	32 (10.1)	
Education, *n* (%)			**0.002**
≤High school	123 (29.6)	129 (40.8)	
>High school	292 (70.4)	187 (59.2)	
Income, *n* (%)			**0.001**
≤$30,000/year	242 (58.3)	223 (70.6)	
>$30,000/year	173 (41.7)	93 (29.4)	
BMI, mean (SD)	28.35 (6.3)	28.79 (6.1)	0.344
Ever smoker, *n* (%)			0.638
Yes	134 (32.3)	96 (30.4)	
No	281 (67.7)	220 (69.6)	
**Exposures**			
WTC dust cloud traumatic exposure, *n* (%)			**0.006**
Yes	190 (45.8)	178 (56.3)	
No	225 (54.2)	138 (43.7)	
Exposure classification, *n* (%)			**0.042**
Worker	212 (51.1)	160 (50.6)	
Resident	82 (19.8)	48 (15.2)	
Clean-up worker	65 (15.0)	73 (23.1)	
Other	56 (13.5)	35 (11.1)	
**Lower respiratory symptoms**			
Cough, *n* (%)			**0.011**
Yes	267 (64.3)	232 (73.4)	
No	148 (35.7)	84 (26.6)	
Wheeze, *n* (%)			**0.002**
Yes	193 (46.5)	185 (58.5)	
No	222 (53.5)	131 (41.5)	
Chest tightness, *n* (%)			**<0.001**
Yes	232 (55.9)	231 (73.1)	
No	183 (44.1)	85 (26.9)	
Dyspnea at rest, *n* (%)			**<0.001**
Yes	144 (34.7)	152 (48.1)	
No	271 (65.3)	164 (51.9)	
**CRP**			
CRP in mg/L, median [IQR] ^b^	1.3 [0.3, 4.5]	1.9 [0.5, 5.8]	**0.004**
CRP > 3 mg/L, *n* (%)			**0.041**
Yes	139 (33.5)	130 (41.1)	
No	276 (66.5)	186 (58.9)	
**WBC**			
WBC in 10^3^ cells/mL, median [IQR]	6.7 [5.6, 8.2]	6.9 [5.8, 8.4]	0.214

^a^ SD, standard deviation; ^b^ IQR, inter-quartile range; ^c^
*p* values were computed based on chi-squared tests for categorical predictors, and two-sample *t*-tests or Mann–Whitney tests for continuous predictors; bold numbers under *p*-value column indicate significant at 0.05 level.

**Table 3 ijerph-19-08622-t003:** Multiple linear regression models for assessing mediation effect of PTSD (PCL score) in the association of WDCTE and CRP level (log(CRP)), adjusted for demographic characteristics and lower respiratory symptoms (*n* = 731).

	Model (1) (Path c)		Model (2) (Path a)		Model (3) (Path b and c’)	
	Log (CRP)		PTSD (PCL Score)		Log (CRP)	
	β	*p*-Value	β	*p*-Value	β	*p*-Value
(Intercept)	−10.95	**<0.01**	43.15	**<0.01**	−11.35	**<0.01**
**PCL score**					0.01	**0.01**
**WDCTE**	0.27	**0.02**	5.06	**<0.01**	0.22	0.06
**Adjusted factors:**						
Sex—Male	−0.20	0.07	−1.78	0.12	−0.18	0.10
Age on 9/11	0.01	0.21	0.001	0.99	0.01	0.21
Race/ethnicity (ref = Hispanic)						
NH-White	−0.14	0.39	−4.04	**0.02**	−0.10	0.53
NH-Black	−0.003	0.98	−5.77	**<0.01**	0.05	0.77
Other	0.09	0.68	−3.98	0.07	0.12	0.55
Education > high school	−0.02	0.87	−1.31	0.35	−0.01	0.95
Income > $30,000/year	−0.04	0.74	−4.56	**<0.01**	0.002	0.99
log(BMI)	3.17	**<0.01**	−0.48	0.87	3.17	**<0.01**
Ever smoker (>1 p-y)	0.44	**<0.01**	−1.16	0.36	0.45	**<0.01**
Exposure category (ref = Clean-up Worker)						
Resident	0.08	0.71	−4.34	0.06	0.08	0.68
Local Worker	0.06	0.75	−1.89	0.33	0.12	0.58
Other	0.12	0.59	−3.92	0.08	0.15	0.48
Lower respiratory symptoms						
Cough	0.33	**0.01**	1.38	0.29	0.32	**0.01**
Wheezing	0.12	0.33	1.35	0.29	0.11	0.38
Chest tightness	−0.08	0.49	4.03	**<0.01**	−0.12	0.32
Dyspnea at rest	0.15	0.21	3.43	**0.01**	0.12	0.32
**Mediation Effect of PCL score**						
	β	95%CI Lower *	95%CI Upper *	*p*-value *		
Total effect (c)	0.27	0.03	0.49	**0.03**		
Direct effect c’)	0.22	−0.02	0.45	0.07		
Mediation effect (c-c’ = a * b)	0.05	0.01	0.09	**0.01**		
Proportion Mediated ((c-c’)/c)	17.0%	2.1%	85%	**0.03**		

* The 95% confidence interval (CI) and *p*-value of mediation effect of PCL score were estimated based on a quasi-Bayesian approximation; path a, b, c, and c’ refer to the paths indicated in Figure 1; bold numbers under P columns indicate significant at 0.05 level.

**Table 4 ijerph-19-08622-t004:** Mediation effect of each PTSD symptom cluster (sub-PCL score) in the association of WDCTE and CRP level (log(CRP)), adjusted for demographic characteristics and lower respiratory symptoms (*n* = 731).

	Re-Experiencing		Avoidance		Negative Cognitions/Mood		Arousal	
	β	*p*-Value	β	*p*-Value	β	*p*-Value	β	*p*-Value
Total effect	0.27	**0.03**	0.27	**0.03**	0.27	**0.03**	0.27	**0.03**
Direct effect	0.22	0.07	0.24	**0.05**	0.23	0.06	0.24	**0.05**
Mediation effect	0.05	**<0.01**	0.03	**0.02**	0.04	**0.02**	0.03	0.12
Proportion Mediated	17.5%	**0.03**	10.1%	**0.05**	13.1%	**0.05**	9.0%	0.14

*p*-values were estimated based on a quasi-Bayesian approximation; bold numbers under *p*-value columns indicate significant at 0.05 level.

## Data Availability

The datasets used in this paper are from the clinical databases at the WTC EHC Data Center. We only used de-identified and anonymized information. The datasets are not publically available, but the de-identified and anonymized information is potentially available upon reasonable request to the WTC EHC Data Center.

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
