# Peer review of "Posttraumatic Stress Disorder Mediates the Association between Traumatic World Trade Center Dust Cloud Exposure and Ongoing Systemic Inflammation in Community Members"

_ijerph, 2022, doi:10.3390/ijerph19148622_

Round 1

Reviewer 1 Report

Quality paper of interest to journal readership - ongoing research project of considerable value and importance to not only the scientific establishment but the people impacted that day, and their families.  While there has work published on the topic, the piece makes particular sense in context of the special issue.

 I didn't find much to critique. It is a competently written paper by clear experts in the field, who have been following this work for a long time.  Their contribution is thus valuable and a worthy addition to the journal.  From a subjective standpoint, I found the literature review and supportive materials to be adequate.

Author Response

The authors appreciate the insightful comments and positive evaluation of our manuscript and would like to thank the reviewer for careful reading of the manuscript.

Reviewer 2 Report

This study aimed to investigate the relationship between systemic inflammation and exposure to World Trade Center dust on September 11, 2001. The study theme fits the aims and scope of this journal and provides readers with useful information, if the study was appropriately conducted. However, there are several issues that need to be addressed, mainly on the methodology of the study.

Major points

  1. Definition of research participants: Are study participants the community residents in the area where WTC dust has been reached out? Does this include residents who moved in after 911? Have the study participants already registered as victims of 911? Since the study is not very large, with 731 patients who visited the hospital, it seems that those who were not actually involved in the WTC should be excluded.

  2. Other factors than WTC dust: The authors specifically focused on WTCdust in this study, but I wonder why they did not include other factors related to exposure to WTC (e.g., presence or absence of injury). Since respiratory is not the only disease that may be related to systemic inflammation, is it not necessary to consider other factors as well?

  3. Exclusion criteria of the study participants: The authors did not mention about the exclusion criteria in the Methods section. For example, it is reasonable to think that people who had an inflammatory disease before 2001 will be excluded.

  4. Ethnicity and social backgrounds of the study participants: The authors explained in the Introduction section that the members of the WTC EHC have diverse age, races and ethnicities, as well as a wide range of social status. Could the authors explain why? Is it possible that the participants in this study reported symptoms related to their traumatic experiences other than WTC? I think it would be good if this point could be noted in the discussion.

Minor points

  1. The sentences in line 91-95 and in 297-300 are almost identical.

  2. Line 132: DSM-V should be “DSM-5”.

  3. Line 315: References are needed for the previous studies on “IL-6 and other cytokines”.

Author Response

This study aimed to investigate the relationship between systemic inflammation and exposure to World Trade Center dust on September 11, 2001. The study theme fits the aims and scope of this journal and provides readers with useful information, if the study was appropriately conducted. However, there are several issues that need to be addressed, mainly on the methodology of the study.

The authors appreciate the comments and inquiries and would like to thank the reviewer for the detailed questions and constructive suggestions. Next, we provide point by point responses to all the questions and suggestions.

Major points

  1. Definition of research participants: Are study participants the community residents in the area where WTC dust has been reached out? Does this include residents who moved in after 911? Have the study participants already registered as victims of 911? Since the study is not very large, with 731 patients who visited the hospital, it seems that those who were not actually involved in the WTC should be excluded.

  1. We appreciate the questions about the study participants and now further clarify the inclusion criteria in the methods section. We state in the beginning of the methods section that the study subjects are enrollees of the WTC EHC at Bellevue Hospital. The WTC EHC is a federally designated treatment and monitoring program for WTC Survivors including local workers, local residents, students and those passing by the area of WTC on 9/11 (https://www.cdc.gov/wtc/eligiblegroups.html#nycSurvivor). All enrollees of the WTC EHC are required to have at least one CDC/NIOSH-certified mental or medical condition related to specific WTC exposures. Some residents who moved in after 9/11 may qualify as “WTC survivors” according to CDC/NIOSH guidelines, e.g., if they moved into the defined geographic area after 9/11 within the defined time frame defined by the CDC/NIOSH WTC Health Program (i.e. for a significant time period before July 31, 2002), and if they have CDC/NIOSH-certifiable mental/medical conditions related to their specific WTC exposures they would qualify to be enrolled in the WTC EHC. All the study subjects are CDC/NIOSH-certified enrollees of the WTC EHC at Bellevue Hospital. It may be worth pointing out that the CDC/NIOSH WTC Health Program (WTCHP) was created to take care of the health of both the WTC Responders and exposed WTC Survivors under the James Zadroga 9/11 Health and Compensation Act passed by US Congress in 2010.  While the WTC involvement and related exposures for WTC Responders are well publicized and well understood, the extent and intensity of the WTC exposure among the WTC survivors are much less well known. We also agree with the reviewer that it is important to clarify the involvement and evidence WTC-related exposures for the study subjects. In response to the last comment, for clarification, we have added, in the Introduction section, the following statements about WTC exposures of WTC Survivors: “In short, acute or chronic 9/11-related exposures for WTC Survivors including children and pregnant women can be quite substantial. In fact, even among children in the WTC-affected community, 12 years after 9/11, we and others have identified increased serum dioxins and furans in children who experienced WTC dust at home [5-14].”

  1. Other factors than WTC dust: The authors specifically focused on WTC dust in this study, but I wonder why they did not include other factors related to exposure to WTC (e.g., presence or absence of injury). Since respiratory is not the only disease that may be related to systemic inflammation, is it not necessary to consider other factors as well?

We appreciate the comments on other comorbid diseases and other exposure factors. We agree with the reviewer that some other factors such as physical injury on 9/11 related to the WTC exposure may also affect systemic inflammation. Very few of our WTC Survivors are “certified” for physical injury on 9/11, thus, there is inadequate sample size to include injury or other possible factors in the statistical analysis. We focus on the WTC dust cloud traumatic exposure (WDCTE) on 9/11. As discussed in the paper, the WTC dust cloud exposure on the day of 9/11 included both exposure to dust/fumes and exposure to traumatic psychological stress. Because PTSD is a main focus of the paper, we focused on the traumatic WTC dust cloud exposure on 9/11. However, we agree with the reviewer that, 20 years after 9/11, further research on long-term health impacts of all factors related to WTC exposures on WTC Survivors is still needed.  In addition to the traumatic WTC dust cloud exposures on 9/11, other exposures (e.g. physical injury) will be investigated in subsequent studies. We believe that the potential effect of these other possible factors has already been accounted for as part of the traumatic experience on 9/11, i.e. they are part of the WTC dust cloud traumatic exposure (WDCTE) and would not affect validity of our study findings. We agree that respiratory is not the only disease that may be related to systemic inflammation, it is important to consider other comorbidities in future studies. In response to this comment, we also added the following statement in the Discussion section: “Third, we adjusted for co-morbid LRS and obesity but future studies should consider randomly sampling patients and then obtain levels of systemic inflammation markers in order to reduce the impact of comorbidities. Respiratory conditions and obesity are not the only types of WTC-related comorbidities associated with inflammation. Thus, it is desirable to adjust for other possible WTC-related co-morbidities in future studies.”

  1. Exclusion criteria of the study participants: The authors did not mention about the exclusion criteria in the Methods section. For example, it is reasonable to think that people who had an inflammatory disease before 2001 will be excluded.

We appreciate the useful comments. We now include a statement about our exclusion criteria in the Method section. We included all patients with CRP measures at the WTC EHC Bellevue Hospital and did not exclude people that satisfy the inclusion criteria. Obesity can be considered an inflammatory condition, and we have included the variable BMI in our assessment. We have also added the following statement in the Discussion section “The current study has multiple limitations. First, the study participants were selected based on the inclusion criteria stated in the method section which are, in turn, based on available patient information. Additional exclusion criteria may be helpful, for example, including the presence of co-existing inflammatory disease before 9/11 and/or who had traumatic experiences other than WTC exposure.” In addition, subsequent studies should include assessment of other systemic inflammatory disorders such as hypertension or diabetes.

  1. Ethnicity and social backgrounds of the study participants: The authors explained in the Introduction section that the members of the WTC EHC have diverse age, races and ethnicities, as well as a wide range of social status. Could the authors explain why? Is it possible that the participants in this study reported symptoms related to their traumatic experiences other than WTC? I think it would be good if this point could be noted in the discussion.

We appreciate the comments and we now more clearly explain the ethnic and social background and the related WTC-related exposure profiles. The cohort of first responders are mainly white male professionals at working age. However, as stated in the paper, the members of our WTC EHC cohort are local community members in NYC who can be either males or females, Hispanics or non-Hispanics, black or white, children or elders etc. Thus, WTC EHC cohort has more diverse ethnicity and social backgrounds compared to responder cohort. We now clarify this in the introduction. Indeed, compared to the general WTC Responders, who are predominantly trained professionals, the members of the WTC EHC are civilians untrained for disasters who were exposed during their normal work, school, or residential activities. The WTC Responders, are predominantly white males, whereas the members of the WTC EHC have equal gender distribution, diverse age, races and ethnicities, as well as a wide range of social economic status. There exist a high percentage of low-income individuals as can be seen in Table 1. The WTC Survivors enrolled at the WTC EHC also have different exposure profiles than the WTC Responders. We also agree that it would be desirable to know the information whether the participant had symptoms related to traumatic experiences other than WTC, but the available clinical databases generally do not have this kind of information. This type of information is useful and can be obtained for random subsamples of the participants at WTC EHC via future research projects. We have indicated this limitation in the discussion. In particular, we added the following: “The current study has multiple limitations. First, the study participants were selected based on the inclusion criteria stated in the Method section which are, in turn, based on available patient information from existing clinical databases. Additional exclusion criteria may be helpful for future studies, for example, including the presence of co-existing inflammatory disease before 9/11 and/or who had traumatic experiences other than WTC exposure.”

Minor points

  1. The sentences in line 91-95 and in 297-300 are almost identical.

We thank the reviewer for the suggestion. The sentences have been revised.

  1. Line 132: DSM-V should be “DSM-5”.

We thank the reviewer for the suggestion and we have corrected to DSM-5 in the text.

  1. Line 315: References are needed for the previous studies on “IL-6 and other cytokines”.

We thank the reviewer for the suggestion and we have added the following references about IL-6 and other cytokines:

Kim, T. D., Lee, S., & Yoon, S. (2020). Inflammation in post-traumatic stress disorder (PTSD): a review of potential correlates of PTSD with a neurological perspective. Antioxidants, 9(2), 107.

Passos, I. C., Vasconcelos-Moreno, M. P., Costa, L. G., Kunz, M., Brietzke, E., Quevedo, J., ... & Kauer-Sant'Anna, M. (2015). Inflammatory markers in post-traumatic stress disorder: a systematic review, meta-analysis, and meta-regression. The Lancet Psychiatry, 2(11), 1002-1012.

Del Giudice, M., & Gangestad, S. W. (2018). Rethinking IL-6 and CRP: Why they are more than inflammatory biomarkers, and why it matters. Brain, behavior, and immunity, 70, 61-75.

Reviewer 3 Report

In manuscript ijerph-1805092 a mediation analysis was conducted to investigate the association between acute WTC dust cloud traumatic exposure (WDCTE) on 9/11, chronic PTSD symptoms, and levels of systemic inflammation. The data indicate that the chronic PTSD symptoms and some specific symptom clusters of PTSD significantly mediate the WDCTE on systemic inflammation as reflected by CRP levels. As both chronic PTSD and systemic inflammation are long-term risk factors for neurodegeneration and cognitive decline, further research on the implications of this finding is warranted

The study is interesting and well structured. The introduction is based on a robust foundation of literature. The methods are adequate and adequately described. The results are innovative, well presented, and adequately discussed. The limitations of the study are also stated. The conclusions are interesting and essentially present an interesting advancement of knowledge.

Author Response

(The authors gave the same response as above.)

Round 2

Reviewer 2 Report

The manuscript has been much improved. I appreciate the authors' great efforts to revise the manuscript, especially on clarifying the inclusion and exclusion criteria of the study.